# Diagnostic Test Accuracy of the 4AT for Delirium Detection: A Systematic Review and Meta-Analysis

**DOI:** 10.3390/ijerph17207515

**Published:** 2020-10-15

**Authors:** Eunhye Jeong, Jinkyung Park, Juneyoung Lee

**Affiliations:** 1College of Nursing, Korea University, Seoul 02841, Korea; preaquaty@korea.ac.kr (E.J.); carpe@korea.ac.kr (J.P.); 2Department of Biostatistics, College of Medicine, Korea University, Seoul 02841, Korea

**Keywords:** 4AT, delirium, meta-analysis, sensitivity, specificity, systematic review

## Abstract

Under-recognition of delirium is an international problem. For the early detection of delirium, a feasible and valid screening tool for healthcare professionals is needed. This study aimed to present a scientific reason for using the 4 ‘A’s Test (4AT) through a systematic review and meta-analysis of studies on the diagnostic test accuracy. We systematically searched articles in the EMBASE, MEDLINE, CINAHL, and PsycINFO databases and selected relevant articles on the basis of the predefined inclusion criteria. The quality of the included articles was evaluated using the Quality Assessment of the Diagnostic Accuracy Studies-2 tool. We estimated the pooled values of diagnostic test accuracy by employing the bivariate model and the hierarchical summary receiver operating characteristic (HSROC) model in data synthesis. A total of 3729 patients of 13 studies were included in the analysis. The pooled estimates of sensitivity and specificity of the 4AT were 81.5% (95% confidence interval: 70.7%, 89.0%) and 87.5% (79.5%, 92.7%), respectively. Given the 4AT’s evidence of accuracy and practicality, we suggest healthcare professionals to utilize this tool for routine screening of delirium. However, for detecting delirium in the dementia population, further work is required to evaluate the 4AT with other cut-off points or scoring methods in order for it to be more sensitive and specific.

## 1. Introduction

Delirium is a neuropsychiatric syndrome characterized by acute change and fluctuation of awareness, attention, and cognitive function [1,2]. Delirium in older adults is regarded as a medical emergency due to its high prevalence and a wide range of negative outcomes such as the increased risk of falls, pressure sores, functional decline, higher mortality, and the new onset or deterioration of dementia [3,4]. For this reason, early detection is the key strategy for the management of delirium [5].

Despite a variety of instruments for delirium screening and diagnosis being available, under-recognition by healthcare professionals is still problematic in many care settings [6,7]. For effective detection of delirium, continuous screening embedded in everyday practice is crucially required due to the natural characteristics of the condition presenting acute onset and fluctuating course in a day. Thus, delirium screening tools with both feasibility and accuracy should be used for successful early detection of delirium [8].

According to the recently published delirium guideline, there are several easy-to-use tools for delirium detection that need a short period of time to administrate (<2 min), such as the Simple Question in Delirium (SQiD), modified RASS (m-RASS), and 4 ‘A’s Test (4AT) [9]. Among them, the 4AT has been particularly recommended to use in emergency departments and acute hospital settings, since the tool has been validated and widely used worldwide in those clinical settings. Moreover, the 4AT has the following strengths over other existing tools: no “special” training required, being simple and easy to administer, no physical responses required by patients, all patients can be evaluated (including those untestable due to severe drowsiness or agitation), and the possibility to screen other forms of cognitive impairment due to included brief cognitive tests.

The 4AT consists of four items: (1) alertness, (2) Abbreviated Mental Test-4 (AMT-4), (3) attention (Months Backwards test), and (4) acute change or fluctuating course. Items 1 and 4 are graded 0 (negative) or 4 (positive), while items 2 and 3 are graded 0, 1, or 2, which provides a total score of 0 to 12. The cut-off point is 4, suggesting possible delirium. This means that it reaches cut-off point solely by a single item (1 or 4) since both “altered alertness” and “acute change or fluctuating course” are considered the core features of delirium.

The 4AT has been translated and validated in multiple clinical settings, including acute care hospitals, emergency departments, nursing homes, and geriatric hospitals, internationally [10,11,12]. However, as far as we know, no meta-analysis of diagnostic test accuracy (DTA) of the 4AT for delirium detection has yet been conducted. Therefore, a systematic review and meta-analysis of DTA of the tool are necessary in order to provide the best evidence of the 4AT’s efficacy in clinical settings.

## 2. Materials and Methods

### 2.1. Aims and Design

This study aimed to systematically review and perform a meta-analysis to evaluate the DTA of the 4AT. This study followed the recommended guideline of Cochrane collaboration for systematic reviews of DTA [13] and the Preferred Reporting Items for Systematic Reviews and Meta-Analyses-DTA (PRISMA-DTA) guidelines [14,15].

### 2.2. Search Methods and Eligibility Criteria

The literature was searched in February 2020, in EMBASE, MEDLINE, CINAHL, and PsycINFO databases. To identify relevant reports not included during the search, we also reviewed references. The search was carried out using only 4AT, delirium, and DTA-related terms, not including the terms relevant to patients, reference standard tests, and outcomes for obtaining results with high sensitivity [16]. The term “delirium” was combined with validated search terms of the DTA, such as “sensitivity” and “specificity”.

Two authors (E.J. and J.P.) independently searched, reviewed, and selected the studies, using predefined eligibility criteria. We also identified and reviewed full-texts for studies that met the inclusion criteria. When there were discrepancies, we resolved them through discussion with the third reviewer (J.L.).

The eligibility criteria were set as follows: (1) using the 4AT to detect delirium for identifying DTA of the tool; (2) applying a reference standard to diagnose delirium on the basis of a validated tool or standardized criteria of the Diagnostic and Statistical Manual of Mental Disorders (DSM) III, IV, or V; (3) reporting estimates of DTA including true positive, true negative, false positive, and false negative, or sufficient information to derive them; (4) being written in English, (5) being a prospective study in the general clinical settings. Purely observational studies that were inappropriate to test diagnostic accuracy were excluded.

### 2.3. Quality Assessment

The quality of included studies was assessed with the Quality Assessment of Diagnostic Accuracy Studies-2 (QUADAS-2) tool [17]. The QUADAS-2 is the most used and recommended quality assessment tool for DTA studies. The tool has four domains including “patient selection”, “index test”, “reference standard”, and “flow and timing” [14]. The applicability concerns are evaluated on the basis of the first three domains by identifying if the setting and included patients match the predefined research question.

In this study, a low risk of bias was declared only when all the questions of the tool were answered with “yes”. A high-risk or unclear bias was assigned to the domain if there was at least one answer was either “no” or “unclear”, respectively. Two authors (E.J. and J.P.) independently evaluated the risk of bias and applicability of the included studies, and the third reviewer (J.L.), who is a qualified methodologist of systematic review and meta-analysis, resolved the remaining disagreement.

### 2.4. Data Extraction

The two authors (E.J. and J.P.) independently extracted the data for sensitivity, specificity, and sample size of all included studies. When a study did not report these values but provided sufficient detail for its derivation, we calculated sensitivity and specificity. The following information was extracted from all included studies using a predefined Excel spreadsheet: study characteristics (country, clinical setting, author, and year of publication), sample size, patient characteristics, diagnostic cut-off point, and time taken for administration.

### 2.5. Data Synthesis

On the basis of the recommended guideline of Cochrane collaboration for systematic review (SR) of DTA [13], we planned to employ hierarchical models, which are the most rigorous method to perform a meta-analysis of DTA. Thus, we carried out meta-analysis of DTA studies using two hierarchical models, the bivariate model and the hierarchical summary receiver operating characteristic (HSROC) model [16]. Using these models, we pooled the values for true positives, true negatives, false positives, and false negatives. As further summary measures, we calculated positive likelihood ratio, negative likelihood ratio, and diagnostic odds ratio using the pooled sensitivity and specificity. The results described in the HSROC curve included 95% prediction and 95% confidence regions [18,19].

Moreover, we conducted a pre-planned subgroup analysis based on the quality assessment, namely, subgroup analysis for the studies that have a “low” risk of bias among the four domains of the QUADAS-2. Another post-hoc subgroup analysis with three studies that reported the diagnostic performance of each item [11,20,21] was also performed. Further, we also conducted the sensitivity analysis according to the settings (general wards, emergency department, and stroke unit). The statistical analyses of this study were conducted using R software version 3.2.2 with the package of “mada” (R Foundation for Statistical Computing, Vienna, Austria) [22].

## 3. Results

### 3.1. Search Outcome

Figure 1 presents the details of the study selection flow. Among 1375 records, we identified a total of 1186 studies after removing duplicated articles. Through the screening of titles and abstracts, we identified 70 potentially relevant articles on the diagnostic performance of the 4AT. Among them, we excluded 57 studies, 2 of which were validation studies of the 4AT using the same dataset with already included studies [23,24]. As a result, a total of 13 articles that met the inclusion criteria were finally identified in our systematic review [10,11,12,20,21,25,26,27,28,29,30,31,32].

### 3.2. Study Characteristics

Included studies were conducted in nine different countries and had sample sizes between 49 and 559 participants, comprising a total of 3729 participants. All of the included studies used 4 as the cut-off value of the 4AT for delirium detection. The characteristics of the included studies are summarized in Table 1.

### 3.3. Assessment of Risk of Bias

As a result of the quality assessment, we found nine studies to have a low risk of bias and low applicability concerns in all domains of the QUADAS-2 tool (Table 2). There was no disagreement in quality evaluation between reviewers.

All included studies were evaluated to have a low risk of bias in the domain of “patient selection” except for two studies; one used a case-control design [12], which was categorized as unclear risk of bias, the other, which did not report clear inclusion and exclusion criteria [10], was classified as having a high risk of bias in that domain. Two studies [26,28] were considered to have a high risk of bias in both domains of “index test” and “reference standard test” because these studies used the same tester for two tests without blinding. One study [12] was also assigned as having a high risk of bias for the “reference standard test” domain for having no sufficient information provided in terms of whether the tester was qualified and whether there was blinding in terms of the index test. All studies except two [10,12] used patients receiving the same reference standard, including them in the analysis so that they were regarded as having a low risk of bias in the “flow and timing” domain. The latter two studies did not show a clear distinction in terms of the time intervals between the index test and reference standard test and thus an unclear risk of bias in the domain was assigned. For applicability concerns, none of the studies received anything other than the designation of having a low risk of biases in the “patient selection”, “index test”, and “reference standard test” domains. The included studies in this systematic review could be concluded to have low risk of bias, overall.

### 3.4. Diagnostic Test Accuracy of the 4AT

The diagnostic performance of the 4AT is presented in Table 3. All included studies reported the DTA values of the 4AT including sensitivity and specificity. As a result of meta-analysis, its pooled estimate of sensitivity and specificity were 81.5% (95% CI = 70.7%–89.0%) and 87.5% (CI = 79.5%–92.7%), respectively. For subgroup analysis with nine studies with low risk of bias, we found the pooled sensitivity to be 84.3% (75.4%–90.4%) and that of specificity was 88.5% (79.0%–94.0%). Further, the diagnostic performance of each subtest of the 4AT presented in Table 4.

The results of the sensitivity analysis according to the clinical settings were as follows (pooled sensitivity, specificity, respectively): (1) general wards (78.3% (66.5%–86.8%), 83.5% (76.0%–89.1%)), (2) emergency department (91.6% (83.0%–96.0%), 79.9% (36.7%–96.5%)), and (3) stroke unit (95.3% (86.4%–98.5%), 79.1% (71.6%–85.1%)).

The threshold effect is one of the most important causes of heterogeneity between studies of DTA. If the sensitivity and specificity have an inverse relationship, a coupled forest plot will show a V or an inverted V shape, which represents the fact that there is a threshold effect [33]. Further, when there is a threshold effect, the value of the correlation coefficient between false positive rate and sensitivity will be 0.6 or higher [34,35]. A coupled forest plot of sensitivity and specificity of the 4AT is presented in Figure 2, which confirmed that there seemed to be no threshold effect introduced in our meta-analysis since it was a value of 0.378 and the coupled forest plot was shaped neither as a V nor an inverted V.

The HSROC curve shows a global summary of the test’s diagnostic performance and presents the trade-off between sensitivity and specificity. The HSROC curve in this study had a relatively small confidence region and was positioned in the upper left corner, which supports the desirable diagnostic performance of the 4AT (Figure 3). The overall weighted area under the HSROC curve was 0.91, which also supports at least moderate predictive validity of the tool since it was larger than 0.7.

We also examined an expected positive predictive value (PPV) and a negative predictive value (NPV) for the 4AT across the range of delirium prevalence from 5% to 55%, which was the range reported from the included studies. The best predictive value for the 4AT was observed at 84.7% with a prevalence of about 46% (Figure 4). The result suggests that, when the prevalence is about 46%, the best predictive values of the tool can be achieved. The 4AT also showed relatively high NPV across a wide range of prevalence (low to high) of delirium.

## 4. Discussion

The definition of DTA is the test’s ability to distinguish an incidence or absence of conditions [36]. In order to determine whether a particular tool is beneficial to use in clinical settings, a systematic review and meta-analysis of DTA, which is of paramount importance as scientific evidence of tool effectiveness, should be provided to healthcare providers [18]. The 4AT is one of the most widely used tools for delirium screening internationally [9,37]. Thus far, there has been a systematic review of the tool’s DTA, which includes patients with a particular disease (acute stroke) [38]. This review, however, did not perform a meta-analysis. However, since there have been multiple articles published on the DTA of 4AT in various settings other than stroke units, such as emergency departments, nursing homes, and geriatric hospitals, we argue that it is necessary to evaluate the pooled DTA values of the tool in terms of meta-analysis.

In this study, we used two hierarchical models (the bivariate model and HSROC model), which are the most advanced and rigorous statistical methods to conduct a meta-analysis of DTA by overcoming limitations of the traditional method. The present result of the meta-analysis presented that the sensitivity and specificity of 4AT were 81.5% and 87.5%, respectively, indicating that the 4AT is highly sensitive and specific for delirium detection. Further, we evaluated the risk of bias of studies using QUADAS-2, which is the most recommended quality assessment tool for DTA studies. Our subgroup analysis for studies with a “low” risk of bias based on the QUADAS-2 provided higher pooled sensitivity (84.3%) and specificity (88.5%).

One of the most prominent advantages of the 4AT is that it is simple (<2 min) and no training is required. The Confusion Assessment Method (CAM), as another commonly used tool for delirium detection, requires up to 10 min to administrate [9], and even the short version (Short-CAM) takes longer than 4AT (>2 min). Furthermore, since the range of sensitivity is heterogeneous (46% to 100%) when used routinely for screening purposes, it has been evaluated that special training must be conducted to secure a high DTA of this tool [39]. However, most of the DTA studies of 4AT reported that high DTA levels were achieved without special training.

A post-hoc subgroup analysis with studies reporting diagnostic performance of each item of the 4AT showed that all items were highly specific to delirium. Particularly, “alertness” and “acute change or fluctuating course”, which accounts for items 1 and 4, respectively, are known core features of delirium. For this reason, the tool was designed to use the cut-off point of 4 for items 1 and 4. That is, if a patient is obviously not alert or has symptoms with acute change/fluctuating course, delirium could be suspected. Similarly, our analysis confirmed that both items were highly specific to delirium (item 1 = 97.9%, item 4 = 89.0%). Thus, we could conclude that items 1 and 4 certainly account for securing the specificity of the 4AT for detecting delirium within a high level [8,11].

Disorientation (item 2, AMT-4) and inattention (item 3, Months Backwards test) are symptoms that can occur in cognitive impairment as well as delirium. The results showed that both items 2 and 3 were highly sensitive but less specific for detecting delirium when the cut-off is set at 1 point for each item. However, with more severe deficits (two or more mistakes on the AMT4, or an untestable condition in both items), the specificity was improved; in particular, item 3 (inattention) was highly specific (95.4%). These findings suggest that the severe deficits both in orientation and attention are also useful indicators of delirium. However, the point here is that the patients considered “untestable” on the AMT4 (item 2, 2 points) and Months Backwards test (item 3, 2 points) of the 4AT can also reach cut-off point (4 points) together, which can possibly contribute the increased false positives of the tool. Yet, healthcare professionals should also consider the fact that, to a large degree, such untestable patients (except coma) are more likely to be diagnosed with delirium [40].

This issue was discussed by Richardson et al. [41], who dealt with detection of delirium superimposed on dementia (DSD) using tests for inattention and arousal, in which the sensitivity and specificity of the attention test (90%, 64%) as well as that the arousal test (85%, 82%) were increased when combined together (94%, 92%). The inability to perform simple attention tests alone might not be a useful marker of delirium in the dementia population, but it could reach higher sensitivity and specificity if the core features of delirium are combined. Similarly, detection of DSD may be difficult only with disorientation (item 2) or inattention (item 3) of the 4AT; however, by combining with the key delirium symptoms such as altered alertness (item 1) and acute change (item 4), and applying different optimal cut-offs or scoring mechanisms for this population, the DTA could be improved. Further work will establish if the 4AT with other cut-off points or scoring methods can provide more sensitive and specific measures of delirium in the dementia population.

The 4AT is a tool for screening rather than diagnosis of delirium. The instruction of the tool clearly states that further assessment to reach a diagnosis may be necessary even if cut-off point or more were scored. This tool is a rapid and brief tool for the initial assessment of delirium and cognitive impairment prior to diagnosis. For the tools with primary purposes of screening, an ability to rule out negative cases is clinically more important because the implementation of tailored preventive strategies and further diagnosis for all “possible delirium” is the key factor of delirium care [8]. This implies that specificity and NPV, rather than sensitivity and PPV, are more meaningful measures. The present result confirmed that the 4AT has a high specificity and NPV, by which it can be concluded that the tool is a highly effective screening tool.

The recently published evidence-based guideline recommended using the 4AT over many other tools for delirium detection in emergency departments and acute hospital settings [9]. The results of this study added the best scientific evidence for the DTA of the 4AT and also suggest this tool to be used in routine clinical practice. However, as a result of the sensitivity analysis according to clinical settings in this study, we found that the 4AT has different DTA values depending on the settings, which showed less sensitivity but slightly more specificity in general wards (sensitivity 78.3%, specificity 83.5%) than in emergency departments (91.6% and 79.9%) and stroke units (95.3% and 79.1%). These results suggest that there might be a need to develop a setting-specific tool in order to achieve the DTA, especially in terms of specificity.

Further, as revealed in this study, there is a lack of evidence on the DTA of the tool in intensive care units (ICU), where delirium is commonly observed and has multiple adverse effects on the patients’ prognosis [42,43]. Additionally, the evidence of the possibility to address subsyndromal delirium (SSD) is also limited. SSD is a condition that does not meet the DSM-5 criteria but has one or more features of delirium. It is considered clinically important since it occurs frequently and increases mortality, length of hospital stay, cognitive impairment, and new development of delirium [44,45]. For the wider use of the tool, therefore, more studies on the DTA of the tool should be further carried out, especially in ICU patients, as well as studies on the ability of the 4AT to detect SSD.

## 5. Limitations

Some limitations of the present study should be acknowledged. First, the present results might not free of a publication bias that exaggerates the estimate of DTA, as has been the case in other systematic reviews. Second, the 4AT was used by multiple trained or untrained raters, which makes the assessment of inter-rater reliability necessary, but this was not considered in most included studies. This should be addressed in future studies. Third, the results might be susceptible to an inherent bias because of a threshold effect, which is known as the essential causes of heterogeneity in DTA studies. Yet, the coupled forest plot of sensitivity and specificity of the study showed that there was no evident threshold effect.

Lastly, the quality of this systematic review is dependent on the sample sizes of the included studies and the risk of bias. For this reason, the additional subgroup analysis including only for low-risk bias was also conducted, showing better DTA values. Further work is therefore needed to confirm the performance of the tool on the basis of higher-quality study designs with a larger population for a more expanded application of the tool.

## 6. Conclusions

Our study suggests that 4AT is a valid and feasible delirium detection tool. Given its good diagnostic performance and practicality, it can be considered as an appropriate delirium screening tool, especially for routine use in general wards, emergency departments, and stroke units. Moreover, since this tool covers so-called “untestable” patients for delirium assessment and further intervention, it can be more widely used in clinical settings where those with severe cognitive impairment are common. We, therefore, suggest the use of the tool in more varied clinical settings in which there is a need of a delirium screening tool that has a sufficiently high DTA but where there is a lack of time for using other longer tools or a lack of adequate training to use the tools. Nevertheless, further work is required to evaluate the DTA of 4AT in ICU patients as well as the possibility of 4AT with other cut-off points or scoring methods to be more sensitive and specific measures of detecting DSD and SSD.

## Figures and Tables

**Figure 1 ijerph-17-07515-f001:**
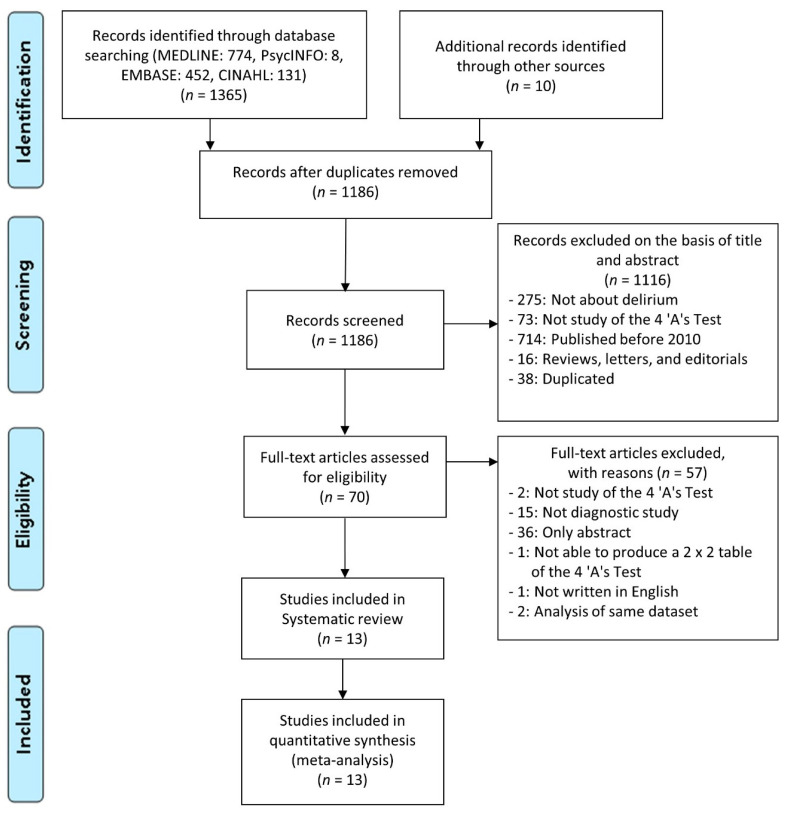
The flow chart of the search for eligible studies.

**Figure 2 ijerph-17-07515-f002:**
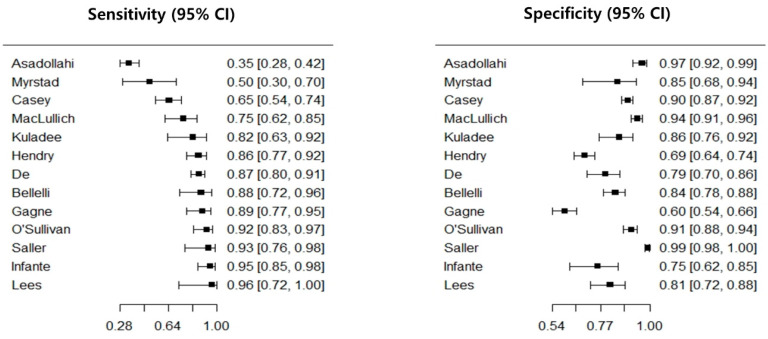
Coupled forest plot of the 4AT. CI, confidence interval; 4AT, 4 ‘A’s Test.

**Figure 3 ijerph-17-07515-f003:**
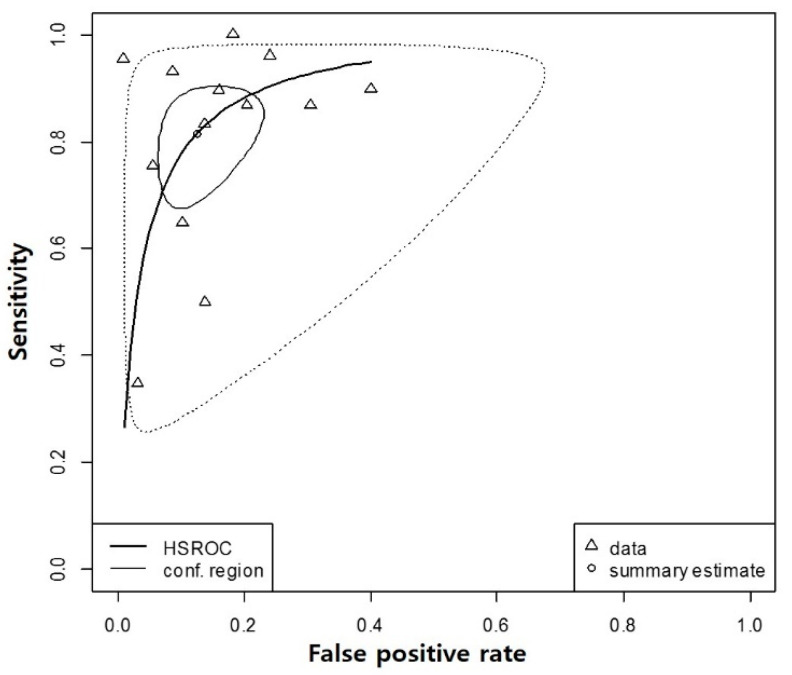
Hierarchical summary receiver operating characteristics curve of the 4AT. HSROC, Hierarchical summary receiver operating characteristics curve; 4AT, 4 ‘A’s Test.

**Figure 4 ijerph-17-07515-f004:**
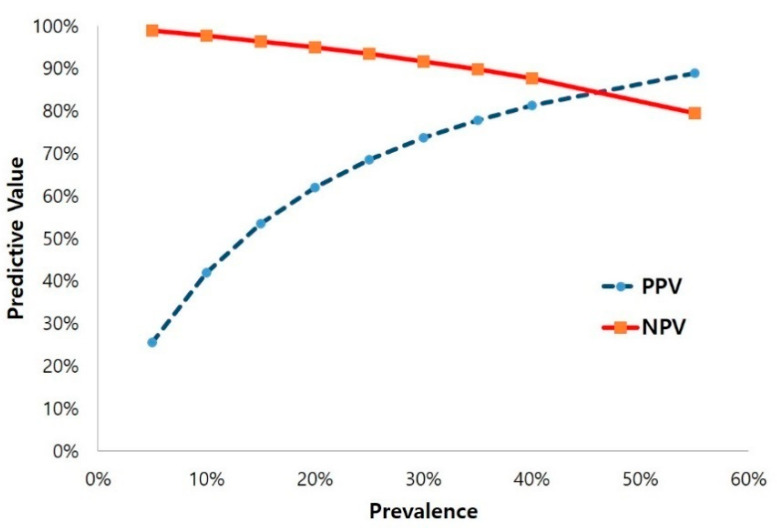
Predictive value of the 4AT. NPV, negative predictive value; PPV, positive predictive value; 4AT, 4 ‘A’s Test.

**Table 1 ijerph-17-07515-t001:** Characteristics of the studies that were systematically reviewed.

First Author	Year	Country	Setting	*n*	Age (M ± SD or Median [Range])	Reference Standard	Cut-off Score	TP	FP	TN	FN	Item Analysis
Asadollahi	2016	Iran	Nursing homes and daily caring centers	293	69.3 ± 1.47	DSM-V	>3	57	4	125	107	Not done
Myrstad	2019	Norway	Acute geriatric ward	49	87 (68–99)	DSM-V	>3	10	4	25	10	Not done
Casey	2019	Australia	Inpatient wards	559	73 ± 16.4	3D-CAM	>3	59	48	420	32	Not done
MacLullich	2019	United Kingdom	ED, medical admission units, MOE units	392	81.4 ± 6.4	DSM-IV	>3	37	19	324	12	Done
Kuladee	2016	Thailand	General medical wards	97	73.6 ± 8.17	DSM-IV, TDRS	>3	20	10	63	4	Done
Hendry	2016	United Kingdom	Geriatric medical assessment unit	434	83.1 ± 6.7	DSM-V	>3	72	107	244	11	Not done
De	2017	Australia	Geriatric and orthogeriatric services	257	86.0 ± 7.3	DSM-V, CAM	>3	138	20	78	21	Not done
Bellelli	2014	Italy	Acute geriatrics ward and department of rehabilitation	236	83.9 ± 6.1	DSM-IV	>3	26	33	174	3	Done
Gagne	2018	Canada	ED	319	76.84 ± 7.4	CAM	>3	44	108	162	5	Not done
O’Sullivan	2018	Ireland	ED	350	77 ^a^	DSM-V	>3	54	25	267	4	Not done
Saller	2019	Germany	PACU	543	52 ± 18	DSM-V, CAM-ICU	>3	21	4	517	1	Not done
Infante	2017	Italy	Stroke unit	100	79 (19–93)	DSM-V	>3	48	12	38	2	Not done
Lees	2013	United Kingdom	Acute stroke unit	100	74 (64–85) ^b^	CAM	>3	12	16	72	0	Not done

CAM, Confusion Assessment Method; CAM-ICU, CAM for the intensive care unit; DSM, Diagnostic and Statistical Manual of Mental Disorders; ED, emergency department; FN, false negative; FP, false positive; M, mean; MOE, medicine of the elderly; *n*, sample size; PACU, post-anesthesia care unit; SD, standard deviation; TDRS, Thai Delirium Rating Scale; TN, true negative; TP, true positive; 3D-CAM, 3-Minute Diagnostic Interview for the CAM; ^a^ median; ^b^ interquartile range.

**Table 2 ijerph-17-07515-t002:** Results of risk of bias assessment of the included studies.

First Author (Year)	Risk of Bias				Applicability Concerns		
	Patient Selection	Index Test	Reference Standards	Flow, Timing	Patient Selection	Index Test	Reference Standard
Asadollahi (2016)	unclear	low	low	unclear	low	low	low
Myrstad (2019)	low	low	low	low	low	low	low
Casey (2019)	high	low	high	unclear	low	low	low
MacLullich (2019)	low	low	low	low	low	low	low
Kuladee (2016)	low	low	low	low	low	low	low
Hendry (2016)	low	low	low	low	low	low	low
De (2017)	low	low	low	low	low	low	low
Bellelli (2014)	low	low	low	low	low	low	low
Gagne (2018)	low	high	high	low	low	low	low
O’Sullivan (2018)	low	low	low	low	low	low	low
Saller (2019)	low	low	low	low	low	low	low
Infante (2017)	low	high	high	low	low	low	low
Lees (2013)	low	low	low	low	low	low	low

**Table 3 ijerph-17-07515-t003:** Diagnostic test accuracy of the included studies.

Author	Year	*n*	Sn (95% CI)	Sp (95% CI)	DOR (95% CI) *	PLR (95% CI) *		NLR (95% CI)
Asadollahi	2016	293	0.35 (0.28–0.42)	0.97 (0.92–0.99)	14.92 (5.52–40.28)	10.07 (3.97–25.55)		0.68 (0.60–0.76)
Myrstad ^b^	2019	49	0.50 (0.30–0.70)	0.85 (0.68–0.94)	5.67 (1.52–21.16)	3.33 (1.29–8.65)		0.59 (0.37–0.93)
Casey	2019	559	0.65 (0.55–0.74)	0.90 (0.87–0.92)	15.87 (9.43–26.72)	6.25 (4.60–8.50)		0.39 (0.30–0.52)
MacLullich ^b^	2019	392	0.75 (0.62–0.85)	0.94 (0.91–0.96)	49.92 (22.74–109.62)	13.23 (8.35–20.96)		0.27 (0.16–0.43)
Kuladee ^b^	2016	97	0.82 (0.63–0.92)	0.86 (0.76–0.92)	27.55 (8.20–92.52)	5.78 (3.21–10.42)		0.21 (0.09–0.49)
Hendry ^b^	2016	434	0.86 (0.77–0.92)	0.70 (0.65–0.74)	14.34 (7.40–27.80)	2.83 (2.36–3.38)		0.20 (0.12–0.34)
De ^b^	2017	257	0.87 (0.80–0.91)	0.79 (0.70–0.86)	24.67 (12.68–47.98)	4.18 (2.83–6.18)		0.17 (0.11–0.25)
Bellelli ^b^	2014	236	0.88 (0.72–0.96)	0.84 (0.78–0.88)	39.44 (12.19–127.63)	5.49 (3.92–7.68)		0.14 (0.05–0.37)
Gagne	2018	319	0.89 (0.77–0.95)	0.60 (0.54–0.66)	12.12 (4.84–30.36)	2.22 (1.87–2.65)		0.18 (0.08–0.41)
O’Sullivan ^b^	2018	350	0.92 (0.83–0.97)	0.91 (0.88–0.94)	127.05 (44.74–360.75)	10.61 (7.27–15.49)		0.08 (0.03–0.20)
Saller ^b^	2019	543	0.94 (0.76–0.99)	0.99 (0.98–1.0)	1648.33 (247.14–10993.70)	108.44 (42.94–273.80)		0.07 (0.01–0.31)
Infante	2017	100	0.95 (0.85–0.99)	0.76 (0.62–0.85)	59.75 (14.41–247.77)	3.88 (2.39–6.31)		0.07 (0.02–0.22)
Lees ^b^	2013	100	0.96 (0.72–1.0)	0.82 (0.72–0.88)	109.85 (6.19–1950.64)	5.19 (3.31–8.13)		0.05 (0.00–0.72)
**Pooled estimates ^a^**								
All included studies		3729	81.5 (70.7–89.0)	87.5 (79.5–92.7)	AUC: 0.911			
Subgroup analysis ^b^		2458	84.3 (75.4–90.4)	88.5 (79.0–94.0)	AUC: 0.918			

AUC, area under the curve; CI, confidence interval; DOR, diagnostic odds ratio; NLR, negative likelihood ratio; PLR, positive likelihood ratio; Sn, sensitivity; Sp, specificity; **^a^** bivariate model; **^b^** nine studies with low risk of bias in all domains of the QUADAS-2 tool; * wide range of confidence interval is due to sparse cell data in each of the study results.

**Table 4 ijerph-17-07515-t004:** Diagnostic test accuracy of each item of the 4AT.

Author	Year	Sample Size	Sn (95% CI)	Sp (95% CI)	DOR (95% CI) *	PLR (95% CI) *	NLR (95% CI)
Item 1. Alertness (cut-off point: 4)							
MacLullich	2019	392	0.31 (0.20–0.45)	0.99 (0.98–1.00)	50.0 (13.78–181.41)	35.0 (10.51–116.54)	0.70 (0.58–0.84)
Kuladee	2016	97	0.38 (0.21–0.57)	0.97 (0.91–0.99)	21.30 (4.17–108.74)	13.69 (3.18–59.0)	0.64 (0.47–0.88)
Bellelli	2014	236	0.52 (0.34–0.69)	0.96 (0.93–0.98)	26.65 (9.66–73.53)	13.38 (6.23–28.76)	0.50 (0.34–0.73)
Pooled estimates ^a^		725	39.6 (26.5–54.4)	97.9 (94.6–99.2)	AUC: 0.810		
Item 2. AMT-4 (cut-off point: 1)							
MacLullich	2019	392	0.63 (0.49–0.75)	0.83 (0.78–0.86)	8.29 (4.35–15.80)	3.68 (2.68–5.04)	0.44 (0.31–0.64)
Kuladee	2016	97	0.96 (0.80–0.99)	0.67 (0.56–0.77)	46.96 (5.98–368.73)	2.92 (2.08–4.09)	0.06 (0.01–0.43)
Bellelli	2014	236	0.97 (0.83–0.99)	0.55 (0.48–0.61)	33.66 (4.50–252.05)	2.13 (1.80–2.51)	0.06 (0.01–0.44)
Pooled estimates ^a^		725	90.4 (58.5–98.4)	69.2 (49.8–83.6)	AUC: 0.832		
Item 2. AMT-4 (cut-off point: 2)							
MacLullich	2019	392	0.41 (0.28–0.55)	0.96 (0.94–0.98)	17.51 (7.91–38.76)	10.77 (5.73–20.24)	0.62 (0.49–0.78)
Kuladee	2016	97	0.88 (0.69–0.96)	0.81 (0.70–0.88)	29.50 (7.70–112.97)	4.56 (2.78–7.48)	0.16 (0.05–0.45)
Bellelli	2014	236	0.90 (0.74–0.96)	0.80 (0.74–0.85)	35.09 (10.12–121.62)	4.53 (3.35–6.11)	0.13 (0.04–0.38)
Pooled estimates ^a^		725	77.2 (39.2–94.7)	88.3 (69.7–96.1)	AUC: 0.908		
Item 3. Attention (cut-off point: 1)							
MacLullich	2019	392	0.71 (0.58–0.82)	0.79 (0.74–0.83)	9.41 (4.81–18.43)	3.40 (2.60–4.46)	0.36 (0.23–0.57)
Kuladee	2016	97	0.96 (0.8–0.99)	0.41 (0.31–0.53)	16.05 (2.05–125.36)	1.63 (1.32–2.01)	0.10 (0.02–0.70)
Bellelli	2014	236	0.93 (0.78–0.98)	0.50 (0.43–0.57)	13.37 (3.10–57.68)	1.85 (1.57–2.19)	0.14 (0.04–0.53)
Pooled estimates ^a^		725	89.9 (68.5–97.3)	58.1 (33.6–79.2)	AUC: 0.821		
Item 3. Attention (cut-off point: 2)							
MacLullich	2019	392	0.31 (0.20–0.45)	0.99 (0.98–1.00)	50.0 (13.78–181.41)	35.0 (10.51–116.54)	0.70 (0.58–0.84)
Kuladee	2016	97	0.50 (0.31–0.69)	0.95 (0.87–0.98)	17.25 (4.76–62.48)	9.13 (3.25–25.65)	0.53 (0.35–0.79)
Bellelli	2014	236	0.86 (0.69–0.95)	0.83 (0.77–0.87)	29.69 (9.74–90.53)	4.96 (3.56–6.90)	0.17 (0.07–0.42)
Pooled estimates ^a^		725	57.6 (23.8–85.6)	95.4 (78.8–99.1)	AUC: 0.892		
Item 4. Acute change or fluctuating course (cut-off point: 4)							
MacLullich	2019	392	0.63 (0.49–0.75)	0.83 (0.78–0.86)	8.29 (4.35–15.80)	3.68 (2.68–5.04)	0.44 (0.31–0.64)
Kuladee	2016	97	0.75 (0.55–0.88)	0.88 (0.78–0.93)	21.33 (6.70–67.90)	6.08 (3.16–11.70)	0.29 (0.14–0.57)
Bellelli	2014	236	0.69 (0.51–0.83)	0.94 (0.90–0.97)	36.11 (13.57–96.13)	11.90 (6.52–21.70)	0.33 (0.19–0.57)
Pooled estimates ^a^		725	68.0 (57.7–76.8)	89.0 (79.7–94.3)	AUC: 0.760		

AUC, area under the curve; CI, confidence interval; DOR, diagnostic odds ratio; NLR, negative likelihood; PLR, positive likelihood ratio; Sn, sensitivity; Sp, specificity; ^a^ bivariate model; * wide range of confidence interval is due to sparse cell data in each of the study results.

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
