# Peer review of "Diagnostic Test Accuracy of the 4AT for Delirium Detection: A Systematic Review and Meta-Analysis"

_ijerph, 2020, doi:10.3390/ijerph17207515_

Round 1

Reviewer 1 Report

Before eventual publication, the following aspects may be considered:

Major:

  • Did the authors register their study with e.g. with PROSPERO (https://www.crd.york.ac.uk/prospero/) or any other register of systematic reviews? This would strengthen the overall quality characteristics of their work, since PROSPERO requires a study protocol, which undergoes a peer review process. All sensitivity analyses have to be defined a priori.
  • In other words: is there a protocol for this SR and MA? If so, authors should mention this protocol and, if available, give a source of link for download or review.
  • Authors should clearly present the R packages and procedures they used. E.G. in Table 1, authors nicely present all relevant numbers (TP, FP, TN, FN) necessary to reproduce their results. When using the R package epiR, sensitivities and specificities slightly differ (rounding to two decimal places?). One example, study #7 in Table 1: De et al. (2017), TP: 138, FP: 20, TN: 78, FN: 21). When reproducing the results in R (Table.7 <- matrix(c(138,21,20, 78), ncol=2) # epi.tests(Table.7, conf.level = 0.95)), results slightly differ from the results presented in Table 3:

> epi.tests(Table.7, conf.level = 0.95)

          Outcome +    Outcome -      Total

Test +          138           20        158

Test -           21           78         99

Total           159           98        257

Point estimates and 95 % CIs:

---------------------------------------------------------

Apparent prevalence                    0.61 (0.55, 0.67)

True prevalence                        0.62 (0.56, 0.68)

Sensitivity                            0.87 (0.81, 0.92)

Specificity                            0.80 (0.70, 0.87)

This may simply be a rounding error, but should be clarified by the authors.

  • It might be interesting to see if the 4AT screening tool shows comparable sensitivity and specificity in different patient populations. If the setting (Table 1) allows, maybe the authors might conduct sensitivity analyses in specific patients groups?

Minor:

  • Please clarify the sentence in line 80: Other types of studies including observational design were excluded. – What is meant with this sentence? Weren’t all studies included in this review observational?!
  • There exists a SR and MA of 4AT performance in the stroke unit (Mansutti et al. BMC Neurology (2019) 19:310)
  • Figure 1: What do the authors mean by ‘Guidelines included in Systematic review (n=13)’?

Author Response

Thank you for providing us an opportunity of revising our manuscript (ID ijerph-955574) entitled “Diagnostic Test Accuracy of the 4AT for Delirium Detection: A Systematic Review and Meta-Analysis”. Reviewer #1’s comments are very insightful that make our manuscript more clear and understandable. Based on the comments, we made point-by-point responses to all of them, and associated modifications to the manuscript. These changes all appear in the revised version of the manuscript in red with 12-point Times New Roman. Please see the attachment

Reviewer 2 Report

I read with great interest the mentioned manuscript. This study aimed to present a scientific reason for using the 4AT through a systematic review and meta-analysis of studies on the diagnostic test accuracy. The topic is of interest. The article is very well written, with an impeccable grammatic and an elegant static analysis. However, the manuscript suffers from some shortcomings. My specific comments are below:

Major comments

Abstract

  • P1; L12: Review “For the early detection of 10 delirium, a feasible and valid screening tool for routine use by healthcare professionals is needed” > I disagree with this statement. There are already valid tools for the diagnosis of delirium, with high sensitivity and specificity.
  • P1; L20: Review “Given its best evidence of accuracy and practicality, we recommend healthcare professionals to utilize this tool for routine screening of delirium. Further work is required to evaluate 4AT with other cut-off points or scoring methods to be more sensitive and specific for detecting delirium in the dementia population”. > It is very assertive and it is the opposite of the following sentence.

Introduction

  • P1; L32: Review “For this reason, early detection is known as the key strategy for the treatment of delirium” > treatment vs prevention.
  • P1; L39: Review “According to the recently published delirium guideline, there are several easy-to-use tools for delirium detection which need a short period of time to administrate (<2min), such as the Simple Question in Delirium (SQiD), modified RASS (m-RASS), and 4AT. Among them, the 4AT was particularly recommended to use across the settings since only 4AT has been validated and widely used across the clinical settings global”. > The reference in question only suggests the use of 4at in patients allocated in emergency room.
  • P2; L43: Review “Moreover, the 4AT has the following strengths over other existing tools: no special training required, simple and easy to administer, no physical responses required by patients, all patients can be evaluated (including those untestable due to severe drowsiness or agitation), and possible to screen other cognitive impairment due to included brief cognitive tests”. > Accord to the 4AT site its easy to learn, but this does not mean that you do not need adequate training for your application
  • P2; L54: The 4AT has been translated and validated in multiple clinical settings internationally > but not in ICU patients, like the CAM-ICU and ICDSC

Results

  • P11; L190: “The result suggests that, when the prevalence is about 46%, the best predictive values of the tool can be achieved. The 4AT also showed relatively high NPV across the wide range of prevalence (low to high) of delirium” > And when the prevalence is low, like young patients, admitted in the ward, without neurologic diseases....what is the NPV and PPV of the method and its real utility?

Discussion

  • P11; L198: “The 4AT is the most recommended tool in several guidelines and is widely used for delirium screening internationally” > I strongly disagree with this sentence.
  • P11; L215: “A recently published study comparing DTA of the CAM and 4AT reported that the sensitivity of CAM (40.0%) was much lower than 4AT (76.0%), although training was conducted only 216 about the CAM” > A series of studies published in the last decade report much superior sensitivity of CAM-ICU. I think it is an extremely biased sentence.
  • P12; L225: “Thus, we could conclude that items 1 and 4 certainly account for securing the specificity of the 4AT for detecting delirium within a high level” > however 4at does not address the burden of delirium, as well as the possibility of subsyndromic delirium.
  • P12; L229: “The results showed that, both items 2 and 3 were highly sensitive but less specific for detecting delirium, when the cut-off is set at 1 point for each item” > The selected studies excluded patients with previous neurological diseases or in use of sedatives?
  • P12; L238: “This issue can be discussed by Richardson, et al. [40] that dealt with detection of delirium superimposed on dementia (DSD) using tests for inattention and arousal, in which the sensitivity and specificity of the attention test (90%, 64%) as well as that the arousal test (85%, 82%) were increased when combined together (94%, 92%)”. > However, it would lose the meaning of the use of 4AT, which would be a quick test and without the need of specific training for its application.
  • P13; L16: “it can be considered as an appropriate delirium screening 275 tool especially for routine use in clinical practice”. > For selected patients;

Tables

  • In Table 3 and 4 the DOR and PLR do not have a very long interval to have statistical significance?

Minor comments

  • See the attached archive

Author Response

Thank you for providing us an opportunity of revising our manuscript (ID ijerph-955574) entitled “Diagnostic Test Accuracy of the 4AT for Delirium Detection: A Systematic Review and Meta-Analysis”. Reviewer #2’s comments are very insightful that make our manuscript more clear and understandable. Based on the comments, we made point-by-point responses to all of them, and associated modifications to the manuscript. These changes all appear in the revised version of the manuscript in red with 12-point Times New Roman. Please see the attachment

Reviewer 3 Report

The methodological approach it's correct, with search in more than three engines and the literature analysis by two independent researchers with discrepancies resolved by a third one. 

The tables and figures are adequate for the study purpose.

Considering that can occur a lot of subjectivity in the assessment of delirium and, that in the diagnostic odds ratios in the studies included in the analysis there is a big interval (Table 3), I propose that instead a recommendation for use 4AT for delirium detection, should be a suggestion, as a good tool for delirium screening.   

In clinical practice there are many situation where occurs disagreement between the practitioners. Examples are the interpretation o pulmonary artery occlusion pressure (Komadina et al, Chest 1991;100:1647-1654) or interventions based on  PAC data (Jain et al, Intensive Care Med 2003;29:2059-2062). I suggested to the authors to modify for recommendation to suggestion in the use of this tool for delirium assessment. Because of this, may be the authors should explore in the comments, the interobserver variability as a limitation to be considered.

Author Response

Thank you for providing us an opportunity of revising our manuscript (ID ijerph-955574) entitled “Diagnostic Test Accuracy of the 4AT for Delirium Detection: A Systematic Review and Meta-Analysis”. Reviewer #3’s comments are very insightful that make our manuscript more clear and understandable. Based on the comments, we made point-by-point responses to all of them, and associated modifications to the manuscript. These changes all appear in the revised version of the manuscript in red with 12-point Times New Roman. Please see the attachment
